# Neonatal Morphine Results in Long-Lasting Alterations to the Gut Microbiome in Adolescence and Adulthood in a Murine Model

**DOI:** 10.3390/pharmaceutics14091879

**Published:** 2022-09-06

**Authors:** Danielle Antoine, Praveen Kumar Singh, Junyi Tao, Sabita Roy

**Affiliations:** 1Department of Neuroscience, Miller School of Medicine, University of Miami, Miami, FL 33136, USA; 2Department of Surgery, Miller School of Medicine, University of Miami, Miami, FL 33136, USA

**Keywords:** pain management, neonates, gut microbiome

## Abstract

Despite the many advancements in the field of pain management, the use of intravenous opioids, such as morphine, in neonates is still a challenge for clinicians and researchers, as the available evidence concerning the long-term consequences of such an early exposure is limited. In particular, little is known concerning the long-term consequences of neonatal morphine exposure on the gut microbiome, which has been identified as a key modulator of health and diseases. Consequently, the purpose of this study was to investigate those long-term consequences of neonatal morphine on the gut microbiome. Newborn mice were exposed to either morphine (5 mg/kg/day) or saline for a duration of 7 ± 2 days. Fecal samples were collected during adolescence and adulthood to longitudinally assess the gut microbiome. DNA extracted from the stool samples were sent out for 16s rRNA sequencing. During adolescence, neonatal morphine resulted in a significant increase of α-diversity and an overall decrease in the abundance of several commensal genera. During adulthood, β-diversity revealed a significantly different microbial composition of the neonatally morphine-exposed mice than that of the controls. The results demonstrate that morphine exposure during this critical developmental period resulted in long-lasting changes, particularly a reduction in several commensal bacteria. Thus, an adjunct therapeutic intervention with probiotics could potentially be used along with opioids to manage pain while attenuating the long-term co-morbidities of neonatal morphine later in life.

## 1. Introduction

Opioid analgesics, such as morphine, are frequently used in the United States for pain management. However, they have serious side effects which limit their use. Currently, the general Centers for Disease Control and Prevention (CDC) guidelines on opioid usage for analgesia do not include recommendations for the neonatal population, as the available evidence concerning the benefits and harms of opioid therapy in neonates is limited [1]. In 2020, 1 in every 10 babies were born prematurely, and received care in the Neonatal Intensive Care Unit (NICU) [2]. During that time, those neonates received painful procedures that required analgesia. The current analgesic treatment in the NICU comprises non-opioids such as acetaminophen and oral sucrose, weak opioids like Tramadol, and strong opioids such as fentanyl and morphine [3]. Therefore, neonates in need of invasive procedures, such as surgery, are exposed to opioids post-natally for pain management and stabilization, even though the known long-term effects of such an early exposure are limited. In adults, side effects of opioid use include but are not limited to addiction, analgesic tolerance, hyperalgesia, and gastrointestinal symptoms [4]. Opioid receptors are very abundant in the GI tract; therefore, opioid administration results in various gastrointestinal side effects, such as gastroesophageal reflux, nausea, vomiting, and constipation [4].

Over the last decade, the gut microbiome has been found to have a significant role in modulating health and diseases. The gut microbiome consists of trillions of microorganisms of thousands different of distinct species, along with their genes, [5] which not only include bacteria but also fungi, parasites, and viruses [5]. A healthy gut microbiome is stable and functions to metabolize barely digestible polysaccharides, detoxify toxic products, and serve as a barrier against pathogens [6]. Overall, the symbiotic relationship between commensal microbiota and the host achieves a balanced and mutually beneficial state. More importantly, numerous signaling molecules derived from gut microbiota, such as metabolites, neurotransmitters, and neuromodulators, can act on the pain receptors and regulate pain signaling [7]. Upon administration, morphine is metabolized through glucuronidation transforming it to morphine 3-glucuronide (M3G) and morphine 6-glucuronide (M6G) in the liver [8]. M6G and M3G are hydrolyzed by β-glucuronidase which is synthesized by both intestinal mucosal cells and gut bacteria [8].

However, morphine administration results in serious negative side effects on the gut microbiome. In adult mice, our lab has previously shown that morphine exposure causes microbial dysbiosis, which is a disruption in the symbiotic relationship between the host and the gut microbiota [9]. Gut microbial dysbiosis can result in a decrease of commensal bacteria, an increase of pathogenic bacteria, or a loss of overall microbial diversity [10]. Furthermore, morphine-induced microbial dysbiosis can result in the disruption of the epithelium barrier of the gut [11,12]. The gut epithelium is one of the most important components of intestinal mucosal immunity, as it prevents organisms in the gut microbiome from invading other organs. Therefore, by compromising the intestinal barrier function, morphine can induce bacterial translocation from the microbiome in the gut to other organs [11], can result in abnormal mucosal immune responses, and can lead to sustained sepsis [13]. Additionally, gut microbial dysbiosis has been implicated in various pain conditions, such as inflammatory pain [14], visceral pain [15] and neuropathic pain [16]. These negative side effects are well known to occur in adults; however, the long-term impact of neonatal morphine on the gut microbiome has not yet been elucidated.

The infant microbiome matures in three phases: a developmental phase; a transitional phase; and a stable phase. The developmental phase, from months 0 to 14, occurs shortly after birth during lactation; the transitional phase, from months 15 to 30, occurs during the weaning period with the introduction of solid foods and the continuation of breast milk feedings; and the stable phase occurs from months 31 to 46 [17]. The early post-natal period is the most dynamic stage of intestinal microbiota development [6]. Thus, during the development phase, the gut microbiome can easily be influenced by various factors, such as morphine exposure, which will dictate the assembly of the commensal ecosystem. Consequently, in this study, we explore the long-term effects of neonatal morphine exposure on the gut microbiome in a murine model.

## 2. Materials and Methods

### 2.1. Experimental Animals

Female and male C57BL/6J mice were obtained from Jackson Laboratory. They were 8–10 weeks old at the time of purchase. A maximum of five mice were housed per cage. Cages were regularly changed and each treatment group was separately handled to avoid contamination between their microbiomes. Food and tap water were available ad libitum. All animals were maintained in a specific pathogen-free facility which was maintained on a 12-h light/dark cycle. All procedures were reviewed and approved by the University of Miami Institutional Animal Care and Use Committee (IACUC) on 20 October 2021. The IACUC protocol number was 20-100-ad01. All procedures were conducted in line with the guidelines set forth by the National Institutes of Health Guide for the Care and Use of Laboratory Animals.

### 2.2. Experimental Design

Nulliparous females, C57/BL6, aged 10–12 weeks, were mated to healthy males of the same age. Four female mice were housed with one male mice. Females were assessed for pregnancy and housed individually upon determination of pregnancy. Upon delivery, pups were randomly assigned to a morphine group (morphine injections) or a control group (saline injections). At this time point, post-natal day 1 was recorded, and all pups were exposed to morphine or saline starting on post-natal day 7 ± 2 days for a duration of 7 ± 2 days total. Pups in the morphine group were administered morphine, 5 mg/kg/day, subcutaneously once-a-day at a volume of 0.01 m^3^. A low 5 mg/kg concentration was used, as 3 mg/kg was found to be the lowest dose to provide analgesia in newborn rats [18]. Similarly, pups in the saline group were administered saline subcutaneously once-per-day at a volume of 0.01 m^3^. During this injection period, pups were placed on a heat pad to ensure animals received proper warmth after separation from their mothers. After the neonatal morphine exposure, pups were left with their mothers, sexed and weaned at 21 days. Fecal samples were collected during adolescence at 4–5 weeks old and adulthood at 8–9 weeks old and sent out for 16s rRNA sequencing.

### 2.3. 16S rRNA Gene Sequencing

DNA was extracted from the collected stool samples using a DNeasy 96 PowerSoil Pro QIAcube HT Kit, along with the QIAcube HT liquid-handling machine (Qiagen, Maryland, USA). The extracted samples were sent to the University of Minnesota and the sequencing was performed by the University of Minnesota Genomics Center. The hypervariable V4 region of 16S rRNA gene was used for PCR amplification using the forward primer 515F (GTGCCAFCMGCCGCGGTAA) and reverse primer 806R (GGACTACHVGGGTWTCTAAT). Utilizing the Illumina MiSeq v.3 platform, the amplicons were sequenced and generated 300-bp paired-end reads.

### 2.4. Bioinformatic Analysis

Amplicon sequence variants (ASVs) were generated with the DADA2 package [19]. The ASV files with abundance values from the V4 region were used for analyses. For diversity analyses and taxonomic assignment, the QIIME2 [20] pipeline was used. Linear Discriminant Analysis Effect Size (LEfSe) [21] was used to detect differentially enriched taxa across groups and the threshold for discriminative features on the Linear Discriminant Analysis score was set to 2.

### 2.5. Statistical Analysis

To analyze the α diversity, the Kruskal–Wallis’s test was used to detect if α-diversity differed between the neonatally morphine- or saline-exposed offspring. Permutational multivariate analysis of variance (PERMANOVA) tests were used to analyze β-diversity and assess if the microbial communities differed between the treatment groups.

## 3. Results

### 3.1. Impact of Neonatal Morphine Exposure on Gut Microbial Diversity

#### 3.1.1. In Adolescence

To investigate the long-term effects of neonatal morphine exposure on the gut microbiome, stool samples from adolescents and adults, both females and males, that were neonatally exposed to morphine or saline were collected. DNA extracted from these samples were sent out for 16s rRNA sequencing. After quality control, unique amplicon sequence variants (ASVs) were identified among the samples. To assess α-diversity, the Chao1 index measuring taxa richness and the Shannon index measuring both taxa richness and evenness were used. Both indexes revealed α-diversity to be significantly higher in the neonatally morphine-exposed adolescents compared with the saline-exposed adolescents (Chao1, *p* = 0.01258; Shannon, *p* = 0.01573) (Figure 1A). Consequently, neonatal morphine resulted in an increase of α-diversity in adolescence at 4–5 weeks old. To assess the β-diversity, the Bray-Curtis dissimilarity index, which measures similarity or dissimilarity of different microbial compositions, was used. The principal-coordinate analysis (PCoA) plot was used to visualize similarities or dissimilarities of the data. Although distinct clustering was observed between the fecal samples of the neonatally morphine-exposed adolescents and the neonatally saline-exposed adolescents, the difference did not reach a significant level (PERMANOVA: F = 1.9425, R^2^ = 0.028, *p* = 0.078) (Figure 1C).

#### 3.1.2. In Adulthood

In adulthood at 8–9 weeks old, both the Chao1 and Shannon indexes revealed no significant difference in α-diversity in the fecal samples of the neonatally morphine-exposed adults compared to the saline-exposed adults (Figure 1B). Contrarily, in adolescence, the PCoA plot using Bray-Curtis distance showed distinct and significant clustering of the neonatally morphine-exposed adults compared to the neonatally saline-exposed adults (PERMANOVA: F = 7.0502, R^2^ = 0.1408, *p* = 0.001) (Figure 1C). Consequently, the microbial composition of fecal samples from neonatally morphine-exposed adults was significantly different from the neonatally saline-exposed adults.

### 3.2. Impact of Neonatal Morphine Exposure on Gut Microbial Composition

#### 3.2.1. In Adolescence

In an effort to pinpoint any sex-dependent effects of neonatal morphine exposure on the gut microbiome, we investigated the composition of the gut microbiome separately in females and in males. Analysis looking at the gut microbial composition revealed a reduction in the relative abundance of Bacteroidetes, Verrucomicrobia and Actinobacteria, as well as an increase in the relative abundance of Firmicutes at the phylum level in the neonatally morphine-exposed female adolescents compared to saline-exposed female adolescents (Figure 2A). With Firmicutes and Bacteroidetes, the two most abundant phyla, such abundances indicated an increase in the Firmicutes/Bacteroidetes (F/B) ratio, as Firmicutes increased while Bacteroidetes decreased. In the morphine-exposed males, however, the analysis showed no change in the relative abundance of the different phyla when compared to neonatally saline-exposed males (Figure 2A).

To further assess the gut microbiome, LEfSe was used to identify differentially enriched microbial taxa from the phylum to the genus levels. An increase in the bacteria from the genera Allobaculum (phylum Firmicutes) was observed in the fecal samples of the neonatally morphine-exposed adolescents compared to the neonatally saline-exposed adolescents (Figure 2B). Additionally, an overall decrease in the relative abundance of bacteria from several genera, including Lactobacillus (phylum Firmicutes), Turicibacter (phylum Firmicutes), Akkermansia (phylum Verrucomicrobiota) and Bifidobacterium (phylum Actinobacteria), was observed in the fecal samples of the neonatally morphine-exposed adolescents compared to the neonatally saline-exposed adolescents (Figure 2B).

Such an overall decrease of bacteria from several genera, following neonatal morphine exposure, was observed in both male and female morphine-exposed adolescents. In females, the decrease in the relative abundance of bacteria included that of the genera Akkermansia (phylum Verrucomicrobiota), Bifidobacterium (phylum Actinobacteria) and Turicibacter (phylum Firmicutes) (Figure 2C). In males, the relative decrease in abundance included bacteria in the genera Lactobacillus (phylum Firmicutes) and Bifidobacterium (phylum Actinobacteria) (Figure 2C).

#### 3.2.2. In Adulthood

Similar to adolescence findings, a reduction in the relative abundance of Bacteroidetes and Verrucomicrobia, as well as an increase in the relative abundance of Firmicutes at the phylum level, were observed in the neonatally morphine-exposed female adults when compared to the saline-exposed female adults (Figure 3A). Consequently, the increase in the F/B ratio persisted in adulthood for female mice. A similar reduction in the relative abundance of Bacteroidetes and Verrucomicrobia and an increase in the relative abundance of Firmicutes at the phylum level was observed in morphine-exposed male adults (Figure 3A). Such abundances were not present in morphine-exposed male adolescents, thus suggesting that the increase in the F/B ratio appeared later in life for male mice.

LEfSe revealed that some of the changes observed in adolescence persisted in adulthood. The increase in bacteria from the genera Allobaculum (phylum Firmicutes) persisted in the fecal samples of the neonatally morphine-exposed adults when compared to the saline-exposed adults (Figure 3B). Similarly, the decrease in the relative abundance of bacteria from the genera Akkermansia (phylum Verrucomicrobiota) and Bifidobacterium (phylum Actinobacteria) persisted in adulthood (Figure 3B). However, some changes were only seen in adulthood, such as an increase in the relative abundance of bacteria from the genera Lactobacillus (phylum Firmicutes) and Turicibacter (phylum Firmicutes) in the fecal samples of the neonatally morphine-exposed adults compared to the saline-exposed adults (Figure 3B). The overall increase of bacteria from the genera Allobaculum (phylum Firmicutes), Lactobacillus (phylum Firmicutes), and Turicibacter (phylum Firmicutes) and the decrease in the relative abundance of bacteria from the genera Akkermansia (phylum Verrucomicrobiota) occurred in both female (Figure 3C) and male morphine-exposed adults (Figure 3C).

## 4. Discussion

The current study shows that brief neonatal morphine exposure results in long-term alterations to the gut microbiome observed in adolescence and adulthood. In a murine model, we show that neonatal morphine induces significant changes to the gut microbial diversity and its composition. In both female and male offspring at all ages, the four phyla, *Firmicutes*, *Bacteroidetes*, *Actinobacteria* and *Verrucomicrobia,* dominated the fecal microbiome. Importantly, we show a consistent increase in the *Firmicutes*/*Bacteroidetes* (F/B) ratio in the neonatally morphine-exposed mice. An increase in the F/B ratio has been associated with gut dysbiosis in the context of obesity, as various studies have found obese animals and subjects to exhibit increased abundances of *Firmicutes* at the expense of *Bacteroidetes* [22,23,24]. Overall, a normal F/B ratio is broadly considered a key factor in maintaining normal intestinal homeostasis. 

Our results further reveal neonatal morphine to have a significant impact on the diversity of the neonatally morphine-exposed adolescents’ microbiome. Investigations of the effects of opioids administered neonatally on the gut microbiome are very limited. However, previous studies looking at the impact of prenatal opioid exposure where opioids were administered to the mothers have shown an impact on α-diversity in the offspring. Using a 60-day chronic methadone treatment, prenatal opioid exposure was shown to increase α-diversity in the offspring at 3 weeks old [25]. Additionally, using a 3-day brief hydromorphone treatment, prenatal opioid exposure was shown to increase α-diversity in the offspring at 5 weeks old [26]. Similarly, in adolescence at 4–5 weeks old, our studies show that neonatal morphine resulted in a higher richness and evenness when compared to controls as measured by both the Chao1 and Shannon indexes. However, such an increase in α-diversity was not observed in adulthood at 8–9 weeks old, as both the Chao1 and Shannon indexes revealed no significant difference between the neonatally morphine-exposed adults and neonatally saline-exposed adults. Consequently, our findings indicate that neonatal morphine causes a significant growth of the microbiota early on, but that significant microbiota growth does not persist into adulthood as the gut microbiome reaches a stable state. A higher α-diversity has historically and widely been associated with beneficial outcomes [27], and prior work looking at chronic morphine treatment in adult mice have reported either no change or a decrease in α-diversity [12,28]. However, prior studies have also reported increased α-diversity in the context of chronic opioid use in adults. Indeed, an analysis of the intestinal microbiota of 45 adult patients with substance use disorders (SUDs), most of whom reported heroin use, and of 48 healthy controls (HCs), revealed an increase in α-diversity [29]. The authors showed bacterial diversity in the SUDs was higher than that in the HCs, and gradually increased with the length of substance abuse [29]. Another study showed increased α-diversity following fentanyl intravenous self-administration in male rats that self-administered at a concentration of 1.25 µg/kg/injection [30]. Consequently, an increase in microbial diversity may in fact be linked to gut microbial dysbiosis and pathology, particularly SUDs.

The adverse or favorable effects of the microbiome on the host depend not only on the quantity of different species, but also on which exact species that are present. The β and LEfSe analysis in our study revealed neonatal morphine to result in significant differences in the bacterial taxa constituting the gut microbiome. In adolescence, there was an overall decrease in the abundance of several commensal genera in the neonatally morphine-exposed mice relative to the saline-exposed controls. In female adolescents, the decrease in the relative abundance of commensal bacteria included that of the genera *Akkermansia*, *Bifidobacterium* and *Turicibacter*, while in male adolescents the decrease in relative abundance included commensal bacteria of the genera *Lactobacillus* and *Bifidobacterium*. A lower abundance of commensal bacteria, particularly a decrease in short chain fatty acid synthesizing communities such as *Lactobacillus* and *Bifidobacterium*, is overall detrimental, and has been associated with adverse outcomes. Indeed, the loss of gut commensals such as *Lactobacillus murinus* and the overgrowth of pathogenic bacteria such as *Klebsiella pneumoniae* have been found to predispose neonates to immunosuppression, increased risk of bloodstream infection, and late-onset sepsis [31]. In general, septic patients were found to have a significant decrease in microbial α-diversity, loss of commensal bacteria, overgrowth of pathogenic bacteria, and a distinct intestinal microbial community compared to those without sepsis assessed by β-diversity [32,33]. Consequently, important implications of a persisting gut microbial dysbiosis following neonatal morphine may involve immunosuppression and increase vulnerability to sepsis later in life.

Notably, *Akkermansia muciniphila,* a specific bacterium of the genus *Akkermansia,* which contributes to the maintenance of a healthy gut barrier, regulates immunity, and limits the onset of inflammation [34]. The lack or decreased abundance of this commensal bacterium is linked with multiple diseases, such as obesity, diabetes, liver steatosis and inflammation [35,36]. The genus *Lactobacillus* is another diverse group that includes many species used in food production and preservation [37]. Several *lactobacilli* in the GI tract have been associated with health benefits, which have prompted their use as probiotics [38]. Bacteria of the genus *Bifidobacterium* also protect the host against pathogens by competitive exclusion [39], and thus provide important health-promoting benefits. Consequently, decreased abundance of these commensal genera indicates an imbalance in the homeostasis of the gut and may potentially expose the host to a higher risk of developing certain diseases. On the other hand, in adulthood, there was an extensive increase in the abundance of several genera in the neonatally morphine-exposed female and male mice when compared to the saline-exposed controls. The overall increase in the relative abundance of some commensal bacteria, such as from the genera *Lactobacillus*, would signal the growth of beneficial microbiota, which might be a sign of microbial recovery. Nevertheless, the persisting decrease in the relative abundance of the commensal bacteria from the genera *Akkermansia* would indicate some level of gut dysbiosis being maintained in adulthood. Consequently, our results indicate neonatal morphine results in persisting changes to the gut microbiome in adolescence—particularly a decrease in several commensal bacteria—with some level of recovery in adulthood as the microbiome stabilizes.

By significantly altering both gut microbial diversity and composition, as well as increasing the F/B ratio, we show that neonatal morphine results in some level of gut microbial dysbiosis in both adolescence and adulthood. Similar to the alterations we found in the gut, several previous pre-clinical studies investigating the long-term effects of neonatal morphine have found other imbalances, particularly neurodevelopmental and behavioral changes. Indeed, morphine administration in neonatal rats from post-natal days 1 to 7 was found to result in persistent weight loss, retarded motor development, and increased anxiety-like behavior in adulthood [40]. Additionally, many brain regions of the adult rats, including motor areas, had decreased metabolic activity, which suggested decreased functional activity in these areas [40]. Moreover, neonatal morphine was also found to cause increased hypersensitivity to pain assessed with the hot plate and tail-flick paradigms [41,42]. Overall, neonatal morphine has been found to result in several negative outcomes in neurodevelopment and behavior. Additionally, in the gut microbiome, several studies have reported morphine to have a direct impact on the gut microbiome composition and result in gut dysbiosis [9,12,43]. However, these studies were conducted in adults and investigated the immediate impact of morphine on the gut microbiome. To date, no other studies have documented the impact of morphine in the neonatal population and how long the gut dysbiosis can persist after discontinuation. Consequently, for the first time, we show neonatal morphine exposure to result in gut microbial dysbiosis which persists later in life. We also show that these changes occur in both male and female mice with minimal sex-dependent differences. Thus, our results provide insight into the negative outcomes for the gut microbiome following early exposure to morphine.

Currently, neonatal morphine is used to reduce the known harmful impact of neonatal pain [44,45,46]; however, accumulating evidence, including our findings, show that neonatal morphine has long-lasting negative effects. Therefore, the implications of our current study for clinical practice include the potential use of probiotics, as our findings showed neonatal morphine to result in a reduction of several commensal bacteria. Several previous studies have found interventions with probiotics to have positive results and good health outcomes. Indeed, an open-labeled study on fecal microbiota transfer in irritable bowel syndrome patients revealed an improvement in abdominal pain with the increase of the relative abundance of *Akkermansia muciniphila* [47]. Additionally, treatment with *Lactobacillus rhamnosus GG* was found to reduce abdominal pain in children with functional GI disorders [48], and a mixture of *Bifidobacterium infantis* with other bacterial strains was found to improve abdominal pain in children with IBS [49]. Consequently, adjunct therapeutic intervention with probiotics could be used along with neonatal morphine to manage neonatal pain but attenuate the long-term co-morbidities of neonatal morphine later in life.

There are several limitations to our findings requiring further investigation. It is known that morphine-induced microbial dysbiosis includes disruption of the gut epithelial barrier, leading to bacterial translocation [11], which was not investigated in this study. In future studies, we will investigate the consequence of persistent microbial dysbiosis on the gut epithelial barrier following neonatal morphine exposure. Furthermore, we only looked at the impact of neonatal morphine on the gut microbiome; however, neonatal morphine exposure has been found to impact multiple other developing pathways and systems [42]. In particular, neonatal morphine was found to result in prolonged pain hypersensitivity manifested by mechanical allodynia and thermal hyperalgesia in adolescence and early adulthood [20,44]. Therefore, future work will seek to implicate the disruptions in the gut microbiome to known effects of neonatal morphine exposure on health outcomes.

## 5. Conclusions

In conclusion, our study shows that neonatal morphine results in gut microbial dysbiosis that persists in adolescence and adulthood. In both female and male mice, neonatal morphine disrupted microbial diversity, increased the F/B ratio, and decreased the abundance of several commensal genera. Our findings indicate that opioid exposure during early development impairs the development of a normal microbiome and leads to an imbalanced gut microbiome later in life. Disruptions in the gut microbiome have been implicated in dysregulated immune function, pain hypersensitivity, and susceptibility to addiction. Thus, neonatal morphine exposure, resulting in long-lasting alterations in the gut microbiome, may influence immunological and neurobehavioral abnormalities later in life and potentially increase vulnerability to several diseases in adolescence and adulthood. Consequently, targeting the gut microbiome therapeutically, using probiotics at the time of morphine administration, may be helpful in alleviating co-morbidities associated with neonatal morphine use for pain management.

## Figures and Tables

**Figure 1 pharmaceutics-14-01879-f001:**
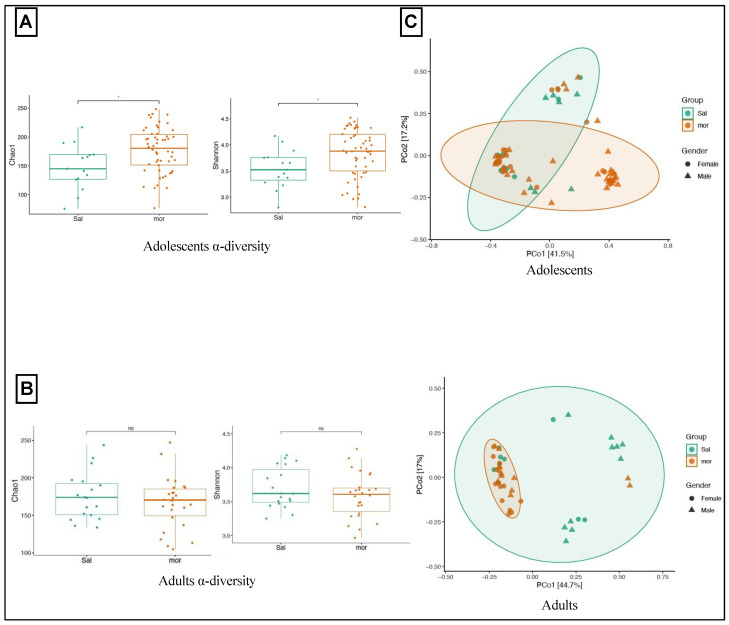
**Microbial diversity analysis.** (**A**) In adolescence, neonatal morphine treatment increased the α-diversity, as measured by the Chao1 (Kruskal-Wallis Rank sum test, * *p* < 0.05; ***right***) and Shannon metrics (Kruskal-Wallis Rank sum test, * *p* < 0.05; ***left***). (**B**) In adulthood, the Chao1 (Kruskal-Wallis Rank sum test, ns: non-significant; ***right***) and Shannon metrics (Kruskal-Wallis Rank sum test, ns: non-significant; ***left***) revealed no significant differences in α-diversity. (**C**) Principal coordinate analysis for β-diversity revealed distinct but statistically non-significant clustering in the neonatally morphine-exposed adolescents compared with the saline-exposed adolescents (PERMANOVA: F = 1.9425, R^2^ = 0.028, *p* = 0.078, ***top***). However, distinct and statistically significant clustering of samples of the neonatally morphine-exposed adults and the neonatally-saline exposed adults were observed (PERMANOVA: F = 2.42, R^2^ = 0.15, *p* = 0.026; ***bottom***).

**Figure 2 pharmaceutics-14-01879-f002:**
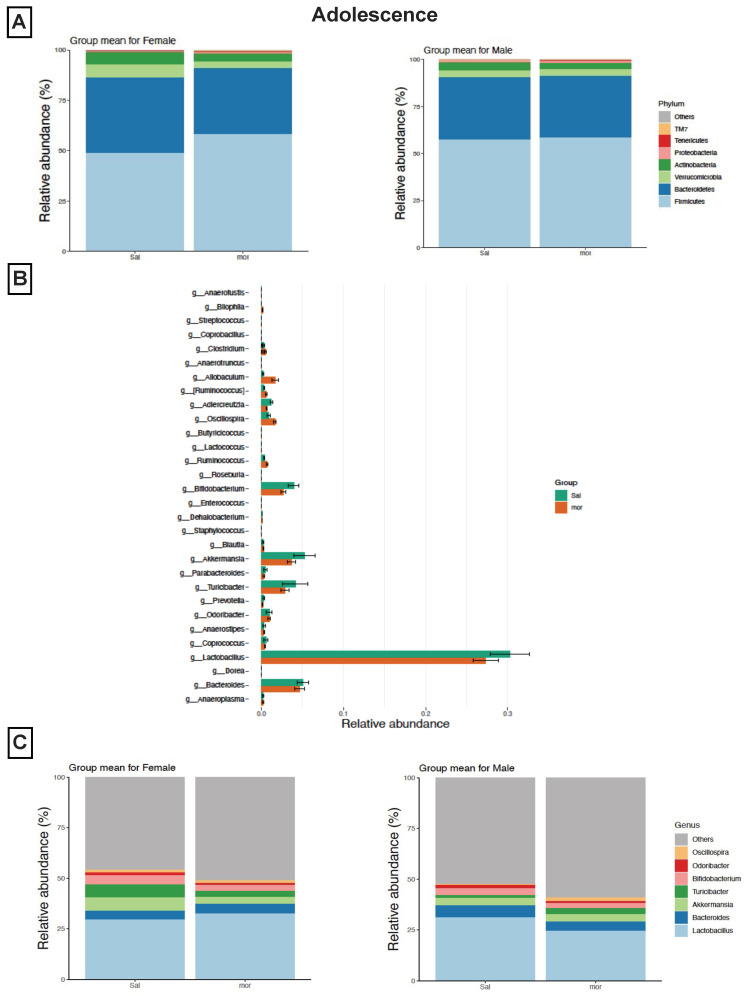
**Microbial composition profiling in adolescence.** Neonatal morphine induces distinct changes in the composition of the gut microbiome in adolescence at 4–5 weeks old. (**A**) Taxonomic distribution of neonatally morphine-exposed and saline-exposed adolescents at the phylum level by sex. (**B**) LEfSe (Linear Discriminant Analysis Effect Size) analysis of bacterial taxa at the genus level among all samples. (**C**) Taxonomic distribution of neonatally morphine-exposed and saline-exposed adolescents at the genus level by sex.

**Figure 3 pharmaceutics-14-01879-f003:**
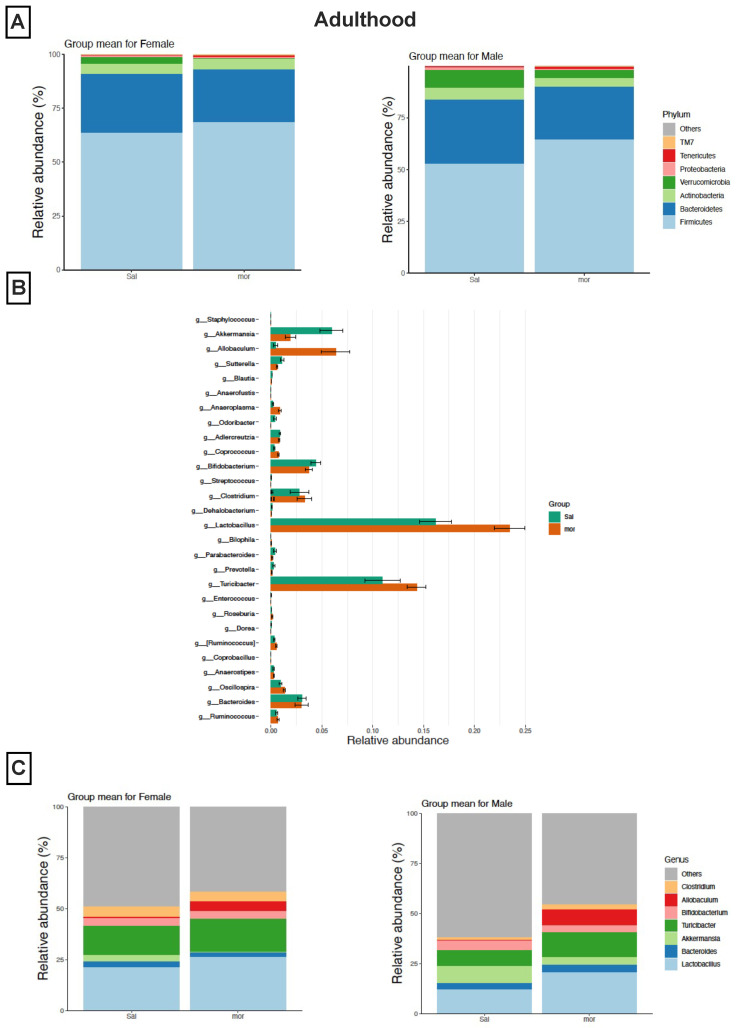
**Microbial composition profiling in adulthood.** Neonatal morphine induces distinct changes in the composition of the gut microbiome in adulthood at 8–9 weeks old. (**A**) Taxonomic distribution of neonatally morphine-exposed and saline-exposed adults at the phylum level by sex. (**B**) LEfSe (Linear Discriminant Analysis Effect Size) analysis of bacterial taxa at the genus level among all samples. (**C**) Taxonomic distribution of neonatally morphine-exposed and saline-exposed adults at the genus level by sex.

## Data Availability

The data presented in this study are available on request from the corresponding author.

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
