# Peer review of "Neonatal Morphine Results in Long-Lasting Alterations to the Gut Microbiome in Adolescence and Adulthood in a Murine Model"

_pharmaceutics, 2022, doi:10.3390/pharmaceutics14091879_

Round 1

Reviewer 1 Report

The manuscript entitled “Neonatal morphine results in long-lasting alterations to the gut 3 microbiome in adolescence and adulthood in a murine model” presents the results, referring to the effects of the neonatal morphine administration on the gut microbiome in adolescence and adulthood mice.

After reading the manuscript, the following doubts and suggestions have arisen.

In the abstract the background, purpose, methods, results, and conclusion should be proportionally and clearly presented. The abstract should be rewritten.

A graphical abstract would be appreciated.

The introduction and the discussion sections should be more complete, providing supplementary background about the theoretical information about the pharmacodynamics effects of opioids in the body, especially on the gastrointestinal tract

The results obtained should be compared with those achieved by other researchers and discussions should be significantly detailed.

In discussion section, the authors need to develop argumentation in depth based on the current understanding and the findings of this study, presenting the potential, the weakness and limitation, and future research direction, among others.

Authors should try to explain the theoretical implication as well as the translational application of their research.

The paper must also contain conclusions regarding the conducted research.

Some other aspects were found in this manuscript:

- the authors should upgrade the references; doi numbers should be provided to each reference.

- spelling check of the text is mandatory.

- English including grammar, style and syntax, should be improved through the professional help from English Editing Company for Scientific Writings.

Reviewer 2 Report

The manuscript titled Neonatal morphine results in long-lasting alterations to the gut microbiome in adolescence and adulthood in a murine model.  the topic is interesting and provides new insights into the implications of the use of neonatal morphine in long-lasting alteration on the gut microbiome. However, there are several points that need to be respected.

In experimental design change 10 ul by international units.

How many days in total was 5 mg/kg/day of morphine administered? It is not clear in the text.

It is convenient to add a section on animals where the author indicates the strain of animals used, housing characteristics, environmental conditions and feeding. No. of a protocol approved by the institutional bioethics committee. where the animals were obtained. Note that both male and female mice were used.

Did Both groups eat and were exposed to the same environmental conditions, free of pathogens? What care did the operator take to avoid contaminating them with their microbiota?

The author said that early opioid use and an increase in microbial diversity may in fact be linked to gut microbial dysbiosis and pathology (which pathologies are reported to relate to this phenomenon?. Does the author suggest that microbial dysbiosis could develop pathologies?

Also, the author report that a lower abundance of commensal bacterial, a decrease in short chain fatty acid synthesizing communities such as lactobacillus and bifabacterum, is overall detrimental and has been associated with several diseases (the author should indicate which condition). There is evidence that a decrease in short chain fatty acid synthesizing develops the disease. Indicate which disease, the author should put attention to pain and inflammation. 

The discussion is very ambiguous. The information provided by the author suggests to the reader that 1) there is not much information to support his findings; 2) What suggests their results? Is evident that morphine in neonatal mice produces gut dysbiosis, but what is it the important and possible implication of the dysbiosis in neonatal mice ; 3) there is no final conclusion that clearly indicates what the contribution of his research is. Although the author suggests how it could impact his research, it is essential to continue researching on this line. The conclusions and implications for clinical practice are left to the reader's interpretation.

Round 2

Reviewer 1 Report

The authors mostly responded to the comments and suggestions and the manuscript was revised accordingly. I consider it could be accepted for publication in this journal.